# Analysis of Running Gait in Children with Cerebral Palsy: Barefoot vs. a New Ankle Foot Orthosis

**DOI:** 10.3390/ijerph192114203

**Published:** 2022-10-30

**Authors:** Federica Camuncoli, Alessia Barbonetti, Luigi Piccinini, Eugenio Di Stanislao, Claudio Corbetta, Gabriele Dell’Orto, Filippo Bertozzi, Manuela Galli

**Affiliations:** 1Department of Electronics Information Technology and Bioengineering, Politecnico di Milano, 20133 Milan, Italy; 2Scientific Institute, IRCCS Eugenio Medea, Bosisio Parini, 23842 Lecco, Italy; 3ITOP SpA Officine Ortopediche, Palestrina, 00036 Rome, Italy; 4Department of Mechanical Engineering, Politecnico di Milano, 20156 Milan, Italy

**Keywords:** running, cerebral palsy, AFO, children, ankle foot orthosis, hemiplegia, sports

## Abstract

Running is an essential activity for children with cerebral palsy (CP). This study aims to characterize the locomotor pattern of running in hemiplegic children with new generation ankle foot orthosis (AFOs) conceived to foster intense motor activities such as running. A group of 18 children with spastic hemiplegia was recruited. A biomechanical multivariable comparison was made between barefoot and with AFO running trials. The focus was devoted to bilateral sagittal plane hip, knee, ankle kinematics and kinetics, and three-dimensional ground reaction forces. Wearing the orthoses, the children were found to reduce cadence and the duration of the stance phase as well as increase the step and stride length. The new AFO resulted in significant changes in kinematics of affected ankle both at initial contact 0–3% GC (*p* < 0.017) and during the entire swing phase 31–100%GC (*p* < 0.001) being the ankle more dorsiflexed with AFO compared to barefoot condition. Ankle power was found to differ significantly both in absorption and generation 5–10%GC (*p* < 0.001); 21–27%GC (*p* < 0.001) with a reduction in both cases when the AFO was worn. No statistical differences were recorded in the GRF components, in the affected ankle torque and hip and knee kinematics and kinetics.

## 1. Introduction

Running is the ability to generate a flight phase consistently between alternating foot strikes [1]. Running is an essential activity in a child’s physical development, both for typically developing (TD) children and for children with pre-existing conditions such as cerebral palsy (CP) [2]. It is also crucial for social interactions since many recreational activities require the execution of this motor activity [3].

Children with CP of GMFCS level I and II (GMFCS: Gross Motor Function Classification System) [4] are usually prescribed ankle foot orthoses (AFO) to facilitate locomotion. The current use of AFO is usually limited to the walking gait, excluding higher impact activities such as running and jumping [5]. It is well documented that the majority of children with cerebral palsy GMFCS level I-II are able to perform running activities [2,6,7,8,9,10,11,12,13,14,15]. Previous studies stated that considering a group of 72 children with unilateral CP, 67% are able to run, but the dynamic balance requirements of running still remain challenging for about 33% of them [16,17]. In this context, the development of custom-made orthotic devices, such as AFOs, may be critical to facilitate running activities while reducing energy cost.

If compared to walking, running requires greater involvement of the musculoskeletal system [2,18,19], often deficient in children affected by neuromuscular diseases. In particular, the most involved muscles in running propulsion are the hip flexors and extensors, hip adductors, knee extensors, and ankle plantar flexors [2,19,20]. The latter are often affected by spasticity, reducing propulsive force application in the limited stance time of running with a consequent reduction in speed [6,21]. AFO designed for sports activities can help overcome these limitations. 

There are different types of AFOs (Figure 1), such as stabilization or proprioceptive type (also known as wDAFO or TRAFO), custom-made, or involving predisposed components. The most common among the stabilization AFOs is the solid AFO (sAFO), typically used in presence of a support equinism. 

Variants of the sAFO with an anterior socket, called ground reaction AFOs (GRAFO) or spiral AFOs, are mainly used in cases of crouch gait. The hinged AFOs (hAFO) may allow to limit and/or elastically control plantarflexion and dorsiflexion. 

Posterior leaf spring (PLS) orthoses, made of either thermoplastic or carbon fiber/kevlar/fiberglass, facilitate dynamic element during the propulsive phase of the stride. 

Despite the variety of AFOs available on the market, the prescription of use is limited to gait only, excluding running. Confirming this, even in the state-of-the-art, only several systematic reviews showed the use of AFOs in children with CP solely during walking [22,23,24]. 

In these studies, sAFOs and hAFOs were compared with barefoot walking, showing an improvement in initial contact, by reducing excessive equinus, and midstance. In addition, it was found that the ankle peak during dorsiflexion was greater while the ankle peak power generation and absorption decreased when wearing sAFO, hAFO, TRAFO, and PLS [15]. 

Pelvis, hip, and knee kinematics did not change when wearing sAFO, hAFO, and PLS. Stride and step length, gait velocity, and single support time increased whereas cadence decreased with sAFO, hAFO, and DAFO when compared to barefoot.

According to Lintanf et al. [23], no extensive study on running of children with cerebral palsy wearing orthotic aids was led, both from a kinematic and dynamic point of view. As stated by Gibson et al. and Chappell et al. [2,25], only few authors including Krätschmer [26], Davis [14], Bohm and Doderlein [8], Kloyia, M. [27] and Iosa [7] have analyzed kinematics and/or kinetics dynamics of barefoot and shoed running in hemiplegic and diplegic children, while Russell investigated the orthoses effect on running on a group of highly trained soldiers [28,29,30]. These previous studies showed that it is possible to modify the locomotor running pattern by moving from rearfoot strike to forefoot initial contact, improving mechanical performance.

This study aims to test a new generation AFO designed to foster beneficial running biomechanical changes in children with CP. The locomotor pattern of running in hemiplegic children has been fully characterized. Specifically, a biomechanical multivariable comparison between barefoot and AFO trials was performed considering hip, knee, ankle joint kinematics and kinetics in the sagittal plane, and the three ground reaction force (GRF) components. 

## 2. Materials and Methods

### 2.1. Orthosis Type

The new AFO belongs to the PLS group, and it is able to result in a greater elasticity than traditional PLS orthoses. This is made possible by combined use of a posterior carbon fiber composite leaf spring, polypropylene, and geometry specifically designed for sports practices. It is featured by a posterior opening to the sandal and a heel cushion made of shock-absorbing material to allow the adipose tissue of the heel expanding during the contact with the ground. In this way, it is possible to absorb the axial stress without traumatizing the lower limb. 

New AFO is an evolution of the Carbon Modular Orthosis (Ca.M.O) [31] an orthoprosthetic custom-made device patented by ITOP (no. Patent: 0001411806), whose peculiarities lie in the combination of different materials, modularity, and proper design of the structural elements (carbon leaf spring, sandal, and calf socket). 

In this study, the new AFO designed in accordance with anthropometric data (foot length, weight, distance between the heel apex, and popliteal fossa) [31] was provided to each child. A pair of orthopaedic shoes for neurological disorders of the same type (they only differ in size and color) manufactured by Duna (Falconara Marittina, Ancona, Italy) were also provided.

### 2.2. Study Design

The research received approval from the ethics committee of the institute “IRCCS Eugenio Medea—Sezione Scientifica Associazione La Nostra Famiglia” and was performed according to the ethical principles set out in the Declaration of Helsinki. All children’s parents or guardians read and signed the informed consent.

Eighteen children with spastic hemiplegia (11 males and 7 females, age: 8.0 ± 1.5 years; weight: 27.4 ± 5.3 kg; height: 129.3 ± 7.3 cm) were recruited for this study (Table 1). The inclusion criteria were as follows: (1) a diagnosis of CP with hemiplegia or limitation prevalent to a lower limb; (2) age between 6 and 11 years; (3) users of AFO orthoses without any time limit; (4) GMFCS level I and II; and (5) Modified Ashworth score [32] less than or equal to 3 on the following muscles groups: triceps surae, hamstrings, and rectus femoris. 

Children using knee and hip tract orthoses (KAFO, HKAFO), having assisted walking needs (use of crutches, walkers, or other), or being uncooperative were not involved in the study. 

Data were collected in the gait analysis laboratory “ASTROLAB” IRCCS Eugenio Medea (Associazione La Nostra Famiglia, Bosisio Parini, Lecco, Italy) equipped with eight-camera optoelectronic system (BTS Smart DX 700 Bioengineering, Milano, Italy) and four coupled force platforms (BTS P-6000, Bioengineering, Milano, Italy). 

Twenty-six passive reflective markers were placed following Davis protocol [33,34]. Specifically, the lower limb segments (pelvis, thighs, shanks, and feet) were used. The trials were performed in two conditions: (1) barefoot running and (2) AFO + orthopaedic shoes running after about 30 days of acclimatization period. 

During tests involving orthopaedic shoes, foot segment markers were placed on the shoe by reproducing as accurately as possible the heel and fifth metatarsal anatomical landmarks. 

Each participant was verbally instructed to run along a straight trajectory of about 12 m from a starting point to a finishing point identified by field delimiters. No indications on running speed or foot impact with the force platforms were given. Due to space constraints, the approach length was 2 m, and the additional distance before and after the region of data collection was 1 m. A sufficient number of trials was performed to have a valid trial in which the child places at least one foot on the force platforms.

### 2.3. Studied Variables 

The spatio-temporal parameters, the ground reaction forces (GRFs), and the kinetic and the kinematic variables at the hip, knee, and ankle joint in the sagittal plane were obtained using a protocol implemented in SMART Analyzer (BTS Bioengineering, Milano, Italy) and post-processed in an ad hoc Matlab^®^ script (MathWorks, Inc., Natick, MA, USA). Once the time instants of initial and final foot contact with the ground were defined, the kinetic and the kinematic data were time normalized to 100 samples and the run cycle with a free float phase was determined in the same way as the gait cycle (%GC). The GRFs were time normalized with respect to the stance phase (i.e., between the contact of one foot on the force platform and the toe off of the same foot). Data from both lower limbs were analyzed to have a valid trial for the more and less affected side. The classifications of the more affected and less affected side were defined with reference to the plegic side shown in Table 1 (column Diagnosis). We considered valid only one trial per child. In case the entire stride for each side was not available in a single trial, one trial per each side (the affected and the unaffected one) was analyzed. 

### 2.4. Statistics 

The statistical analysis was performed through one dimensional statistical parametrical mapping (SPM), using an open-source code (www.spm1d.org) available in Matlab^®^ environment (version R2020a) that uses random field theory to make statistical inferences regarding registered (normalized) sets of 1D measurements [35]. The method is able to identify clusters in the running cycle where the compared variables differ statistically from each other. After testing the normality of the data using the function “spm1d.stats.normality.ttest”, a two-tailed paired *t*-test with Bonferroni correction with a level of significance α = 0.01 (function “spm1d.stats.ttest_paired” implemented in Matlab^®^ environment) was used to compare the two conditions of the trials (barefoot vs AFO). Overall, seven gait cycle variables (Table 2) and twelve kinematic and kinetic variables were statistically tested (Figure 2 and Figure 3).

## 3. Results

All participants were able to comply with the given instructions and run at their self-selected speed with and without wearing the AFO. Participant characteristics are summarized in Table 1. 

The averaged spatio-temporal parameters are reported in Table 2. A statistically significant difference (*p* < 0.010) between the two conditions was found in cadence, stride length, step length, stance, and swing (%GC). No difference was detected in running speed and step width (*p* = 0.120 and *p* = 0.750, respectively).

Wearing the orthoses, the children were found to increase step and stride length as well as reduce both the cadence and the duration of the stance phase, which consequently resulted in an increase of swing phase. The mean and standard deviation of kinematics and kinetics in the sagittal plane of the ankle knee and hip were evaluated. We can see the outcomes both for more and less affected sides (Figure 2 and Figure 3, respectively) in the conditions of barefoot and wearing the new AFO.

The new AFO determined a remarkable (SPM{t}) change in kinematics of the affected ankle both at initial contact 0–3%GC (*p* < 0.017) and during the entire swing phase 31–100%GC (*p* < 0.001) (Figure 2); ankle power significantly differs in absorption 5–10%GC (*p* < 0.001) and in generation 21–27%GC (*p* < 0.001). In AFO condition, it is possible to observe a reduction in the power absorbed and generated (Figure 2). This may be due to the correction of the droop foot during the swing phase along with the fact that AFOs are passive devices which restrict ankle range of motion (ROM).

No significant differences were found In the affected ankle torque and hip and knee kinematics and kinetics (Figure 2). 

Analyzing the results of the less affected side, two under-threshold clusters 0–2%GC (*p* = 0.016) and 89–100%GC (*p* = 0.016) exceeded the critical threshold of 3.803 as the ankle joint during barefoot running was significantly more plantarflexed than in the trials with the AFO (where it is dorsiflexed) (Figure 3). Moreover, a significant difference was detected in the kinematics of less affected hip at the instant of maximum flexion 79–89%GC (*p* = 0.006) (Figure 3). The hip is on average more flexed with AFOs than in the barefoot condition.

No statistical differences were found either in less affected kinetics (hip, knee, ankle torque, and power) and knee kinematics (Figure 3). 

As for the GRFs, Figure 2 and Figure 3 show the averaged values of the three components normalized on the stance phase only. No significant statistical differences were found between trials barefoot and wearing AFO. It is possible to note that the mediolateral less affected side GRF is on average displaced laterally with AFO than in barefoot condition. This means that the foot is more supinated in AFO condition. The averaged vertical GRF component peak is twice the body weight wearing the AFO and 1.8 times the body weight in barefoot condition.

## 4. Discussion

In the present paper, we have focused our attention on the analysis of the ankle joint, since it resulted to be the most affected by AFO. In particular, the joint kinetics and kinematics were evaluated both for the most compromised side and the contralateral one.

According to the results, the running biomechanics of the affected ankle was strongly modified by the presence of AFO. It is possible to observe a reduction both in plantar flexion angle during the swing phase and in the absorption and power generation if compared to barefoot condition. 

As for equinus, considered by Krätschmer et al. [26] to be the first abnormality that needs to be corrected in running, the new AFO seems to be effective in improving it by fostering an initial contact in a more neutral position. 

Results show that AFO may help in reducing the absorbed and the generated ankle power with respect to barefoot condition (affected side). As pointed out by Figueiredo et al. and Lintanf et al. [22,23], this may be the consequence of the use of AFO orthoses since they are non-active devices. They do not generate power at all and only constrain the ankle joint ROM.

It was then observed that AFO seems to correct the kinematics of the compromised side ankle by improving the contact of the foot to the ground. As can be seen from Figure 2, the sagittal plane ankle angle at initial contact in barefoot trial averaged a value of −10 deg, while in the AFO trials the average value increased to 0 deg. This kind of AFO may improve performances and ensure a safer motor activity, not possible with commonly used AFOs.

For what concerns the less affected side, it did not show significant modifications of measured parameters or compensatory mechanisms due to the presence of the orthosis. The mean value of the ankle angle during barefoot and AFO running is basically very similar for most of the gait cycles.

Spatio-temporal parameters (Table 2) were considered to evaluate whether the device improved the motor activity. Decreasing cadence and stance phase duration and increasing step and stride length may be considered as improvements introduced by AFO [8,14].

The obtained results may not be only related to the presence of AFO, but they also could be due to the combined effect of orthosis, orthopaedic shoe, and running velocity. For instance, children might be expected to have performed the tests at a speed they considered safe. In the literature, it is known that shoed running can be featured by a decrease in cadence along with an increase in step length, running speed [36,37] and higher vertical GRF peak values [38,39]. All these findings were also observed in this study. The new AFO, featured by polypropylene calf containment, a foot shell with flexible tip, and a composite posterior leaf spring specifically design to support running activity, may have strengthened the beneficial effects.

The main drawback of this approach may be related to the experimental set up. The presence of four coupled force platforms allowed recording the kinetics of one limb per trial, being the stride length greater than the distance covered by the force platforms. In addition, if on one hand the use of shoe may replicate common life conditions, on the other hand it can also be a limitation for markers placement as well as for the definition of spatio-temporal parameters [36]. During the orthotic trials, the markers were applied to the shoe replicating as much as possible the position of the anatomical landmarks [40]. It was also extremely difficult to analyze the running gait of children with cerebral palsy (CP) wearing only orthopaedic shoes (without the orthosis) due to their size. For this reason, we chose to not acquire these trials in order to minimize the risk of falling and prevent injuries.

Finally, other limitations were the heterogeneity of CP population and the impossibility of having a controlled speed during the trials, since no verbal instructions on running velocity were given. Regarding the limited sample size, the analysis of lot of probably correlated dependent variables (19) from a small sample increased the type II error rate of the study. The possibility of applying Holm’s correction [41] and performing a sample size calculation could be considered for future analysis.

## 5. Conclusions

This study quantified the differences between running with an AFO and barefoot in children with CP and particularly hemiplegia. A new orthosis was specifically designed to allow children with CP to engage in sports activities. We found that the orthosis largely changed the kinematics and kinetics parameters, the prepositioning of the foot, and reduced the power generated and absorbed at the ankle.

Further studies should be carried out to fully understand the benefits that an AFO can bring to the improvement both of running biomechanics and living conditions. This could be done by simultaneously analyzing energy cost and lower limb electromyographic activity. Then, a study to evaluate other movements which occur in daily life such as descending and ascending stairs should be considered for future analysis.

## Figures and Tables

**Figure 1 ijerph-19-14203-f001:**
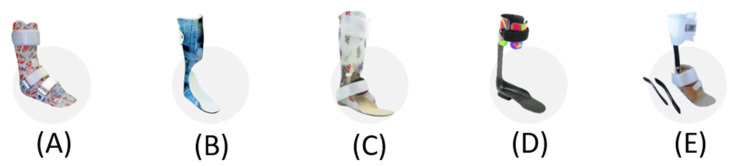
(**A**) solid ankle foot orthosis (sAFO), (**B**) ground reaction AFO (GRAFO), (**C**) hinged AFO (hAFO) (**D**) posterior leaf spring AFO (PLS), (**E**) carbon modular orthosis (CaMO).

**Figure 2 ijerph-19-14203-f002:**
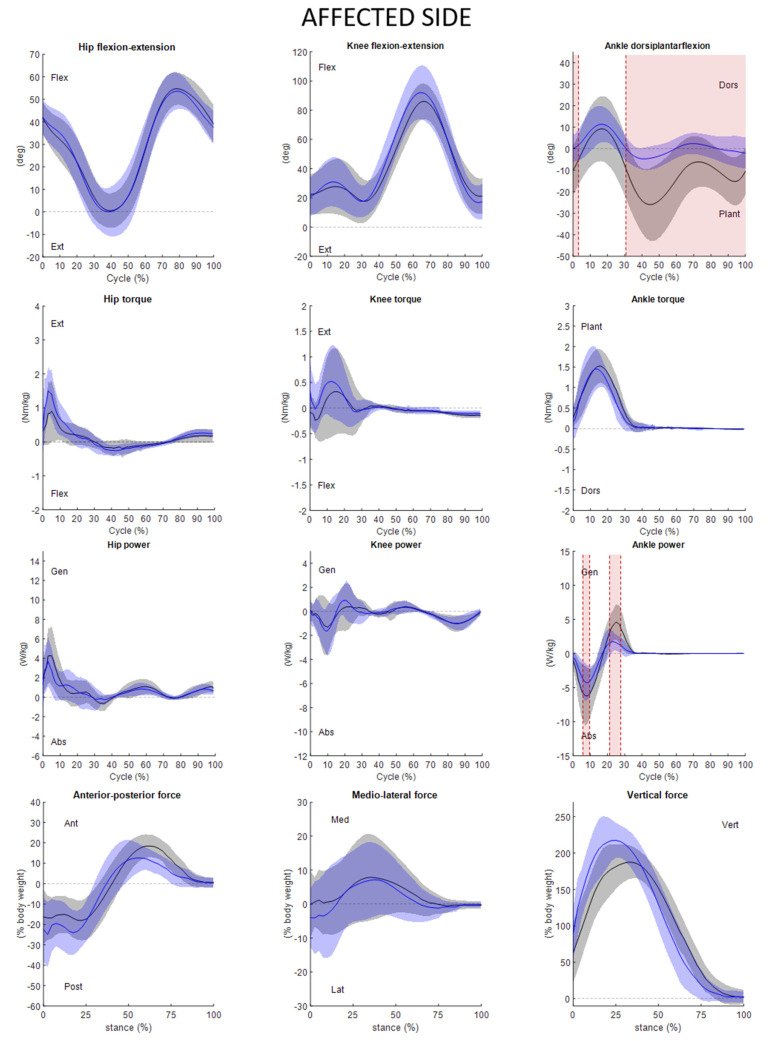
Hip knee ankle kinematic (1st line), torque (2nd line), power (3rd line), and ground reaction forces (4th line) of the affected side. Kinematic and kinetic variables are expressed as a function of the running cycle (Cycle (%) or %GC) while GRFs have been normalized with respect to stance phase (stance (%)). Mean and SD of the barefoot (in black) and the orthotics (in blue) trials are depicted. The SPM statistics results (SPM{t}) are shown in red. Flex = flexion; Ext = extension; Dors = dorsiflexion; Plant = plantarflexion; Gen = generated; Abs = absorbed; Ant = anterior; Pos = posterior; Med = medial; Lat = lateral; Vert = vertical.

**Figure 3 ijerph-19-14203-f003:**
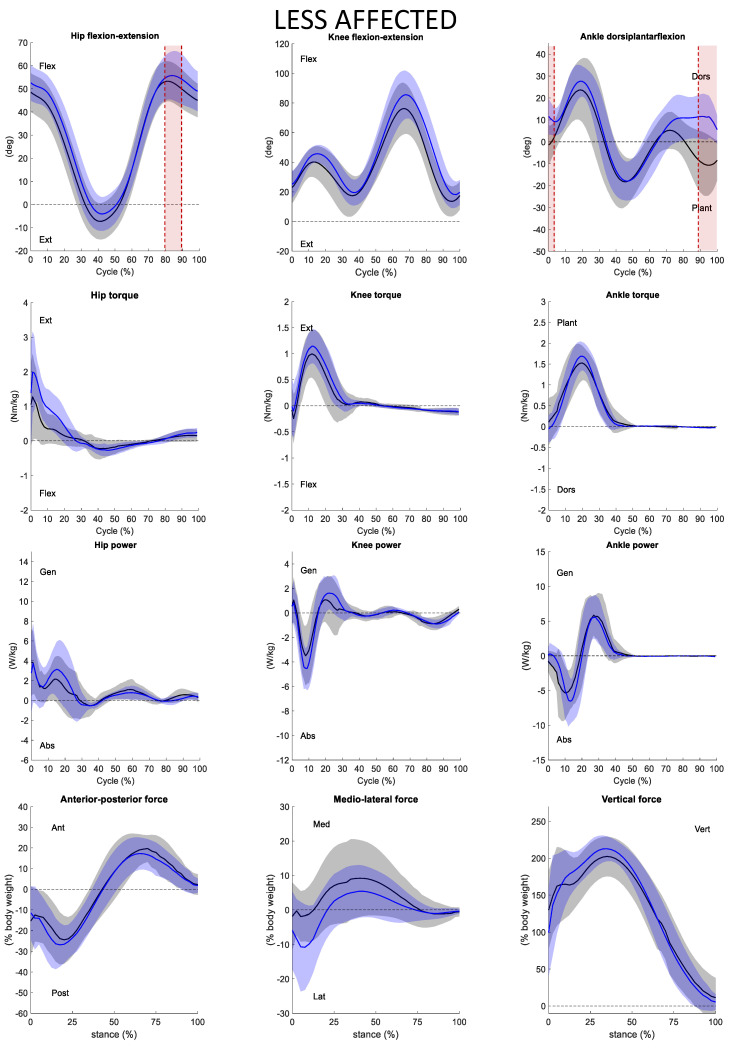
Hip knee ankle kinematic (1st line), torque (2nd line), power (3rd line), and ground reaction forces (4th line) of the less affected side. Kinematic and kinetic variables are expressed as a function of the running cycle (Cycle (%) or %GC) while GRFs have been normalized with respect to stance phase (stance (%)). Mean and SD of the barefoot (in black) and the orthotics (in blue) trials are represented. The SPM statistics results (SPM{t}) are shown in red. Flex = flexion; Ext = extension; Dors = dorsiflexion; Plant = plantarflexion; Gen = generated; Abs = absorbed; Ant = anterior; Pos = posterior; Med = medial; Lat = lateral; Vert = vertical.

**Table 1 ijerph-19-14203-t001:** Participants characteristics. They were classified according to gender, age, weight, height, diagnosis, and GMFCS (Gross Motor Function Classification System).

ID	Sex	Age (Years)	Weight (kg)	Height (cm)	Diagnosis	GMFCS
1	M	9	27.5	137.5	Hemi R	II
2	F	10	39	143	Hemi L	I
3	M	10	23	130	Hemi R	II
4	M	6	23	119	Hemi R	II
5	M	10	32.5	140	Hemi R	I
6	M	10	33	129	Hemi R	I
7	F	7	23	116.5	Hemi R	II
8	M	7	27	135.5	Hemi R	II
9	F	8	31	136.5	Hemi L	I
10	M	8	24	129	Hemi L	II
11	F	7	29	129	Hemi R	I
12	M	10	21.5	130	Hemi L	II
13	M	8	23	124.5	Hemi L	I
14	F	8	37.5	132	Hemi R	II
15	M	7	25	126	Hemi R	I
16	M	7	28	125	Hemi L	II
17	F	5	24	125	Hemi L	I
18	F	7	22	120	Hemi L	II
Mean (SD)	[M: F] 11: 7	8.0 (1.5)	27.4 (5.3)	129.3 (7.3)	[R: L] 10: 8	[I: II] 8: 10

**Table 2 ijerph-19-14203-t002:** Mean and standard deviation (M ± SD) of seven spatio-temporal parameters. Paired *t*-test with Bonferroni correction *p* < 0.010. + values for the affected side. GC: gait cycle.

Variables	Barefoot	Orthosis	*p*-Value
speed (m/s)	2.9 ± 0.5	3.0 ± 0.5	0.120
cadence (step/min)	220.1 ± 0.5	200.4 ± 28.8	**<0.010**
step width (m)	0.11 ± 0.04	0.10 ± 0.05	0.750
stride length (m) +	1.54 ± 0.21	1.75 ± 0.28	**<0.010**
step length (m) +	0.77 ± 0.10	0.86 ± 0.13	**<0.010**
stance (% GC) +	34.7 ± 4.4	31.3 ± 6.2	**<0.010**
swing (%GC) +	65.3 ± 4.4	68.7 ± 6.2	**<0.010**

## Data Availability

Not applicable.

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
