# Peer review of "Analysis of Running Gait in Children with Cerebral Palsy: Barefoot vs. a New Ankle Foot Orthosis"

_ijerph, 2022, doi:10.3390/ijerph192114203_

Round 1
Reviewer 1 Report
INTRODUCTION: There is too much focus on the details about types of available AFO's. This should be summarised into key points relevent to the topic of the research, rather than detailing the specifics of each type of orthoses.
METHODS: Generally well described. Could be improved with grammatical and style editting.
Statistics: I don't understand why only 7 gait cycles were used. Can you also explicitly state that each cycle was defined as "running" ie with a free float phase
RESULTS: The following lines should be moved to discussion, particualrly when the authoirs make h reference to other work: Lines 175-177; and 187-191
FIGURES: Include a key to the abbreviations used in the figures.
TABLE 2. Typo (Variabiles should be Variables) Proved the number for the calculated variables.
DISCUSSION: Suggest some stylistic editting. Line 223 In agreement with [26]" - uses teh authors name rather than the reference number.
Line 233-235 should be in results
General minor typo and stylistic comments for the authors :
Overall the manuscript should be reviewed by an editor for stylistic editing. Some suggestions below but there are many more.
There is overuse of transitional and linking terms (eg However, therefore, in fact, moreover, furthermore etc). Many are not need so could be removed.
Avoid using the terms "suffering from"
Changes from calling subjects within this study "children" interchanging with "persons" and "patients" . Suggest just using one term consitently - prefer "children"
Line 33 Typographical errors " However It", just start sentence with "It..."
Line 37 Change "for some CP patients" to "for some children with CP"
Line 45. "; the role of the AFO" This should be a new sentence
Line 48 ; the most common form -" This should also be anew sentence. Remove excess unnecessar words from teh is sentence " is the one".
Reviewer 2 Report
As the authors highlight, this is one of the first studies on the effect of AFO on the running kinematics in children with CP. However, the authors failed to refer to the study by Buckon et al (2004) who compared the kinematics and kinetics of several types of AFO. The authors of this manuscript describe a range of AFO in their introduction but only include a recently developed AFO in their study and thus should refrain from any conclusion with regard to its superiority compared to other types. Further, there is the limitation of comparing barefoot condition with the shoe and AFO combination. The authors acknowledge this common issue with AFO studies but it does affect the value for clinical practice. Finally, tis manuscript would benefit from substantive English language editing and proofreading.
Specific comments
Abstract
example of lack of proofreading: 'insignificant changes'
Introduction
The authors refer to many studies which include fitness tests for children wit cerebral palsy, although they may be called 'running' test, this does not mean that all participants were able to run.
A definition of 'running' would have been appropriate
Is an AFO an orthopaedic device?
A large section of the introduction is taken up by a description of the different AFOs which are not assessed in this study and thus should be removed or considerably shortened.
Line 70-73: references 1,and 20 are not on children with CP and do not seem to be relevant to the study.
The authors state that the aims was to develop and test but the manuscript only addresses the testing.
I am not sure whether 'multifactorial' is the correct term. What are the different factor in this study?
Methods
How were the participants recruited? What were the inclusion and exclusion criteria?
The methods should also describe how the authors dealt with the marker placement on the shoes. This was only mentioned in the discussion.
'Orthopaedic shoes' should be described in more detail. Did all children wear the same shoes?
Were children told to hit the force plate ? Was there more than one?
How was the single trial selected for analysis. Is one trial sufficient for a valid analysis?
How was normality tested and I assume it was confirmed as the authors used parametric statistics?
Why was a Bonferroni correction used ?
Results
The authors often state statistical significance without stating which direction (i.e. higher or lower in the AFO condition).
It is uncommon to add the % GC together with the statistics between brackets. %GC refers to the outcome measure used not the results of the statistical test.
What us the 'two under-threshold cluster'?
If the authors think that the sample size was too low, they could have included a sample size calculation for future appropriately powered trials.
Please give a reference for Holm's correction.
Conclusions
The conclusions should be improved by adding more detail of the findings and what future studies should focus on.
Round 2
Reviewer 2 Report
Thank you for the reply to my comments.
Please see my reply to your rebuttal (in red)
